# No significant boron in the hydrated mantle of most subducting slabs

Andrew M. McCaig [1], Sofya S. Titarenko[1], Ivan P. Savov[1], Robert A. Cliff[1], David Banks[1], Adrian Boyce[2] & Samuele Agostini [3]

Boron has become the principle proxy for the release of seawater-derived fluids into arc volcanics, linked to cross-arc variations in boron content and isotopic ratio. Because all ocean floor serpentinites so far analysed are strongly enriched in boron, it is generally assumed that if the uppermost slab mantle is hydrated, it will also be enriched in boron. Here we present the first measurements of boron and boron isotopes in fast-spread oceanic gabbros in the Pacific, showing strong take-up of seawater-derived boron during alteration. We show that in one-pass hydration of the upper mantle, as proposed for bend fault serpentinisation, boron will not reach the hydrated slab mantle. Only prolonged hydrothermal circulation, for example in a long-lived transform fault, can add significant boron to the slab mantle. We conclude that hydrated mantle in subducting slabs will only rarely contribute to boron enrichment in arc volcanics, or to deep mantle recycling.

[1] School of Earth and Environment, University of Leeds, Leeds LS2 9JT, UK. [2] Scottish Universities Environmental Research Centre, Rankine Avenue, Scottish Enterprise Technology Park, East Kilbride G75 0QF, UK. [3] Istituto di Geoscienze e Georisorse, Consiglio Nazionale delle Richerche (CNR), Via Moruzzi, 1, 56124, Pisa, Italy. Correspondence and requests for materials should be addressed to A.M.M. (email: a.m.mccaig@leeds.ac.uk)

oron is a key element in tracking the fate of ocean-derived water in subduction zones. Both boron content ([B]) and boron isotope ratio ($\delta^{11}B$) are higher in subduction zone volcanic rocks than in mid-ocean ridge or intraplate basalts, and this is generally seen as a signature of fluids derived from seawater ([B] = 4–5 p.p.m.; $\delta^{11}B$ = +40 ‰) in the melting process. Systematic decreases in B/Be, B/Nb and $\delta^{11}B$ across arcs suggest progressive release of boron from a subducted source rock[1–8], while along-arc variations in B/Zr have been linked to enhanced release of boron from transform faults in the subducted slab[9].

Olivine formed by dehydration of serpentine can contain significant boron[10–12], leading to suggestions that seawater boron may be recycled in slabs into the deep mantle. While this has been rejected on the basis of global volcanic geochemistry[13] recent data on <300 Ma carbonatites suggest addition of isotopically heavy boron to the mantle in the past[14]. Accurate characterisation of the boron chemistry of the altered oceanic lithosphere entering subduction zones is an absolute pre-requisite if boron is to be used as a proxy either in modelling arc volcanism or in deep recycling of volatiles.

Subducted sediments show high [B] but usually negative $\delta^{11}B$[15], and are therefore very unlikely to be the source of positive $\delta^{11}B$ values in volcanics. In contrast, hydration of oceanic crustal and mantle rocks by seawater generally leads to enrichment in [B] and $\delta^{11}B$, in particular serpentinisation of olivine-rich rocks of the upper mantle[16–20]. Dehydration reactions as serpentinite is carried to sub-arc depths lead to release of $^{11}B$-enriched fluid, which is either released from the subducting slab beneath the arc[21,22], or enriches the forearc mantle which is in turn dragged to subarc depths[23,24]. Recent thermal modelling has suggested that conditions along the slab interface, and above the subducting slab, are generally too hot for serpentine to persist to sub-arc depths[25,26], and attention has therefore focused on serpentinisation of the upper slab mantle[6,7,27], in particular during bend-faulting on outer arc rises[28,29].

Increasingly sophisticated models of fluid and mobile component release in subduction zones have been developed[6,7,27,30]. However, the input boron concentration and $\delta^{11}B$ used in these models for subducted slab serpentinites are based almost entirely on samples collected from ophiolitic analogues, or from the seafloor in the Atlantic[16,17,19,20,31], which formed by slow-spreading. These serpentinites are exposed on the seafloor due to large-displacement faulting in fracture zones or oceanic core complexes close to the ridge crest. They cannot be considered typical of the upper mantle in fast-spread lithosphere, which in most cases is relatively intact when entering subduction zones. The overwhelming majority of present day convergent boundaries involve the subduction of fast-spread oceanic lithosphere, and this process dominates additions to the crust through arc volcanics, and potential recycling of volatile elements into the deep mantle. It is therefore critical to constrain the composition of the mantle of fast-spread oceanic lithosphere, but this mantle is inaccessible to direct observation in most cases. In this paper we do not consider past subduction, when the balance of fast- and slow-spread lithosphere being subducted may or may not have been different.

Here we present the first [B] and $\delta^{11}B$ measurements from in situ lower oceanic crust in a fast spreading environment at Hess Deep, in a setting analogous to fault-related alteration in bend faults or transform faults. Both [B] and $\delta^{11}B$ are strongly enriched during alteration of olivine-rich lower crustal gabbros in this setting. We show that sequestration of boron in the lower crust means that it is very unlikely that significant enrichment in [B] or $\delta^{11}B$ occurs in the upper mantle by downward movement of seawater in a bend fault setting. Hence dehydration of slab mantle will not normally be a factor controlling the $\delta^{11}B$

signature of arc volcanics. We also model long term hydrothermal circulation in a large offset normal fault at Hess Deep, showing that this provides a plausible mechanism for introduction of seawater-derived boron below normal Moho depths. This fault is likely to be analogous to an oceanic transform fault in terms of permeability.

## Results

**Hess Deep sample set.** Hess Deep is a propagating rift at the Galapagos triple junction, formed by normal faulting probably because melt supply is insufficient to accommodate spreading. IODP Expedition 345[32,33] drilled several holes towards the base of a ~3 km south-facing fault scarp exposing lower crustal gabbros and upper mantle rocks with an age of 1.27–1.42 Ma[34]. This fault has recently been interpreted as a detachment fault[35]. Three holes at IODP Site U1415 recovered intact sections of layered gabbro including gabbronorite, olivine gabbro and significant intervals of troctolite containing up to 30% olivine. Typical sections include undeformed rocks separated by cataclastic zones with a spacing of 5–30 m. We infer that all samples collected were likely to be in the damage zone of a multistrand fault zone and the alteration intensity is unlikely to be typical of the fast-spread lower oceanic crust away from faults, which remains inaccessible.

The following two styles of alteration are seen in the Hess Deep gabbros. (1) A pseudomorphic background alteration mainly affects olivine, which is replaced by serpentine, saponite, talc, and chlorite (Figs. 1 and 2), with subordinate amphibole, magnetite and sulphides[33,36]. Olivine alteration intensity is normally between 50 and 100%. Variable amounts of prehnite and chlorite affect plagioclase, whereas clinopyroxene and orthopyroxene may be partially replaced by amphibole and talc, respectively. (2) An intense overprinting green alteration is localised along cataclastic zones and veined intervals (Figs. 1 and 2). In these zones, plagioclase is sometimes entirely replaced by prehnite, with minor zoisite, epidote and white mica, and sometimes by mixtures of chlorite and other phyllosilicates (Fig. 2d). Olivine and its alteration products are replaced by chlorite-rich mixed phyllosilicates, with serpentine and saponite being absent in these overprinting alteration zones. Replacement veins of secondary clinopyroxene overprint prehnite and chlorite, and late prehnite and then zeolite veins cut all other assemblages. Cataclastic fault zones are often heavily overprinted by prehnite (Fig. 1b, d). Dykes inferred to be related to Cocos-Nazca spreading[33] are intruded into the cataclastic zones and themselves strongly altered to amphibole and epidote. We infer that the prehnite-chlorite alteration is related to hydrothermal activity during major normal faulting at Hess Deep. The pseudomorphic background alteration might originate in a near-ridge environment, but probably at least partly represents the outer halos of the fault-related hydrothermal alteration.

Table 1 and Fig. 3a show new analyses of 6 whole rock samples for $^{87}Sr/^{86}Sr$, $\delta^{18}O$, [B] and $\delta^{11}B$. The samples range in [B] from 5.8 to 29.4 $\mu g \, g^{-1}$, and in $\delta^{11}B$ from +10.7 to +25.6 ‰. The least altered sample in terms of $^{87}Sr/^{86}Sr$ (AM18) is an olivine gabbro with about 50% pseudomorphic alteration of olivine. It has $^{87}Sr/^{86}Sr$ and $\delta^{18}O$ values close to unaltered ocean crust, and low values of [B] and $\delta^{11}B$. The most altered sample in terms of [B] is the troctolite (AM15), and in the pseudomorphic background alteration, [B] and $\delta^{11}B$ is inferred to correlate with serpentine content (see in situ boron analyses below). Prehnite-chlorite rocks in the overprinting alteration are lower in [B] and $\delta^{11}B$ than the troctolite, and show $^{87}Sr/^{86}Sr$ ratios of up to 0.705, and $\delta^{18}O$ of +4.1 to 5.7‰. Compared with $\delta^{18}O$ fractionation curves[37,38], the data for the most altered samples suggest alteration by seawater-derived fluid at about 200 °C, with secondary clinopyroxene

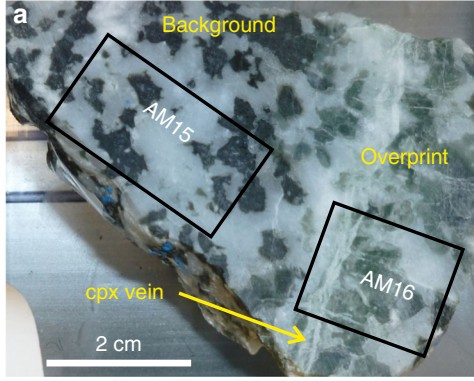

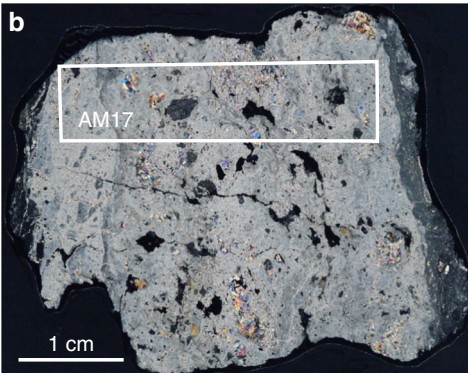

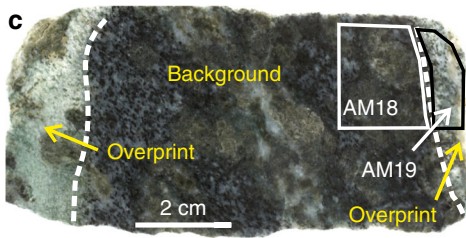

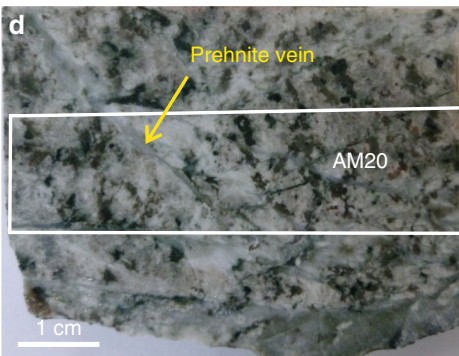

**Fig. 1** Samples analysed in this study. **a** U1415P 23R1 85–90 solid block, showing approximate locations of samples AM15 and AM16 (Table 1). Background alteration (bkgd) overprinted by prehnite-chlorite alteration. Secondary clinopyroxene (cpx) veins are parallel to the alteration boundary, later prehnite veins are perpendicular. **b** U1415J 11R1 44–47. Cataclasite/ breccia, now ~90% overprinted by prehnite, with inclusions of chloritised troctolite and primary clinopyroxene. Dark seams at each side are chlorite-rich cataclasite. Image is thin section, crossed polars. **c** U1415J 8R11 34–137, solid block. Olivine gabbro cut by two seams of overprinting prehnite-chlorite alteration with zeolite veins. **d** U1415I 2R1 25–31, solid block. Cataclastic olivine gabbro with limited prehnite-chlorite alteration, prehnite veins, and some zeolite. At bottom of image is more intense cataclasite overprinted by prehnite

probably indicating an up-temperature hydrothermal overprint, since it can form from prehnite-chlorite assemblages at temperatures above ~ 360 °C[33].

The only previous data for [B] and $\delta^{11}B$ in oceanic gabbros comes from the Oman and Troodos ophiolites[39,40], and from ODP Hole 735B[41] on the Southwest Indian Ridge. Oman data (Fig. 3) overlaps with our data in $\delta^{11}B$, but all our samples are significantly higher in [B], and two samples are significantly higher in $\delta^{11}B$ than any Oman sample. The other datasets are lower in both [B] and $\delta^{11}B$ than ours. In contrast, our altered gabbros do fall within the range of serpentinised and talc-altered peridotites from the mid-Atlantic Ridge, with lower values of [B] most likely reflecting the maximum modal olivine in the protolith of ~30%. Serpentinites exposed on the seafloor at Hess Deep are also enriched in boron with a similar range in [B] to our samples (5–30 µg g$^{-1}$)[20].

Our overall estimate for the composition of fault-altered lower gabbro at Hess Deep is [B] = 17.2 ± 10.7 µg g$^{-1}$, $\delta^{11}B$ = + 21.7 ± 1.9‰ (Table 1). This contrasts with estimates for the whole (unfaulted) gabbro layer in Oman of [B] = 1.3 µg g$^{-1}$ and $\delta^{11}B$ = + 12‰[39].

**Mineral data and the repositories of boron**. Electron microprobe analyses (Fig. 2d and Supplementary Data 1) show that phyllosilicates replacing olivine are mixtures of serpentine, chlorite, saponite and talc. The background alteration sampled in AM15 and AM18 is mainly mixtures of serpentine (lizardite and chrysotile with little antigorite[36]) and saponite, with more chlorite-rich mixtures near olivine-plagioclase contacts. Plagioclase is variably replaced by prehnite, chlorite, white mica and other calcsilicates. In the overprinting green alteration, chlorite dominates all the phyllosilicate assemblages after olivine, and plagioclase is either replaced by chlorite-rich or prehnite-rich assemblages (Fig. 2c). With the exception of lower Cr contents, secondary clinopyroxene is very similar to primary clinopyroxene.

In situ boron contents have been measured using LA-ICPMS, mostly on mixtures of secondary minerals. Data on mixtures should be regarded as semi-quantitative owing to uncertainty in SiO$_2$ content (see "Methods" section). The highest values (36–149 µg g$^{-1}$) are found in serpentine-rich mixtures with variable amounts of saponite, and low Al$_2$O$_3$ indicating limited chlorite contents (Fig. 3d). Chlorite-rich mixtures have lower boron contents (0–26 µg g$^{-1}$) and [B] in late chlorite veins is mostly < 2 µg g$^{-1}$ (our LA-ICPMS detection limit), as it is in late prehnite veins, and in primary olivine, clinopyroxene and plagioclase. Prehnite-rich assemblages replacing plagioclase contain significant boron (10–32 µg g$^{-1}$), and secondary clinopyroxene veins contain small but significant quantities (0–8 µg g$^{-1}$). These data are consistent with the whole rock analyses. The greatest boron contents are in the background-altered troctolite AM15, where olivine is about 60% replaced by serpentine and saponite, and plagioclase about 50% replaced by prehnite and chlorite. The lowest [B] in the background alteration is found in olivine gabbro AM18, which only contained about 10% primary olivine which is mainly altered to saponite. Plagioclase is little altered in this rock. The main boron-bearing mineral in the overprinting green alteration is prehnite, with hydrothermal secondary clinopyroxene, chlorite and late prehnite veins all low in [B]. Of the overprinting alteration samples, AM17 is about 90% prehnite and contains 18 µg g$^{-1}$ of boron, AM16 about 40% prehnite, and AM19 is dominated by a zeolite vein and contains 10–20% prehnite.

An unusual feature of our whole rock data is the apparent correlation of [B] and $\delta^{11}B$, which is not seen in any of the other

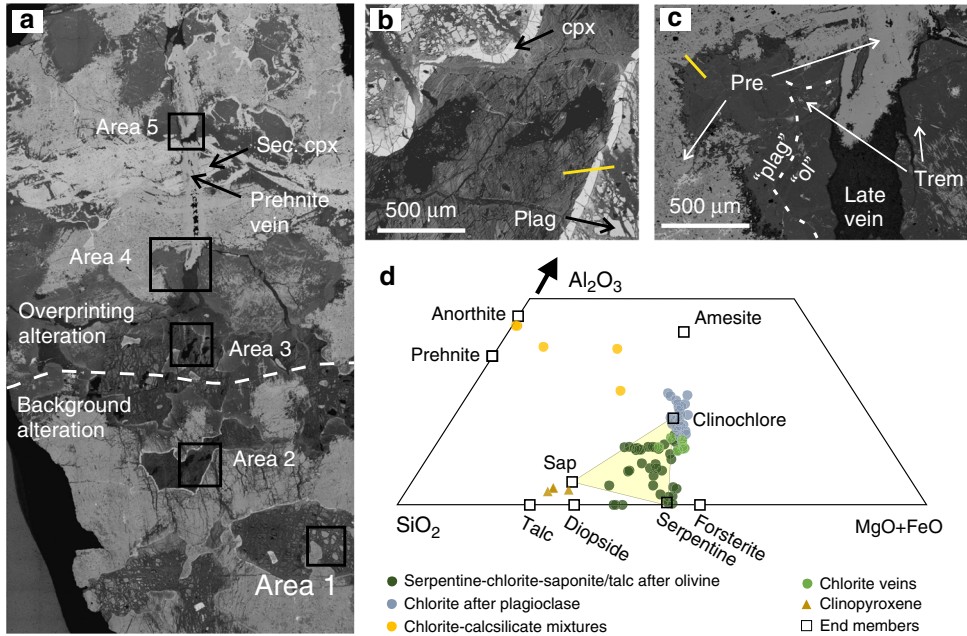

**Fig. 2** Textures and electron microprobe data. **a** Backscatter SEM montage of slide 65984 (U1415P 23R1 85–90, see Fig. 1a) showing the contact between background (AM15) and overprinting (AM16) alteration, and secondary clinopyroxene (cpx) veins cut by late composite veins of prehnite and mixed phyllosilicates. **b** Area 2 shows relict igneous texture, with unaltered clinopyroxene rims separating plagioclase partially altered to chlorite from olivine completely altered to phyllosilicates. Yellow line indicates probe traverse (see Supplementary Data 1 for analyses). **c** Area 4: Igneous minerals are completely replaced apart from minor clinopyroxene. Areas of former olivine are inferred from the presence of tremolite needles formed in an earlier, higher temperature corona reaction between olivine and plagioclase[36,83]. Prehnite (pre) is present both as a complete replacement of plagioclase, and in late veins. **d** Phyllosilicate EPMA data, calculated assuming all Fe is FeO (Supplementary Data 1). Background replacement of olivine in areas 1 and 2 is a variable mixture of chlorite, serpentine and saponite, with some talc. Serpentine disappears almost completely in the overprinting alteration, with the typical mesh texture veins no longer visible. Background replacement of plagioclase is mainly prehnite and other calcsilicates, and this intensifies in the overprinting alteration, where aluminous chlorite or amesite-rich mixtures also replace plagioclase. Veins are mainly chlorite where they cut former olivine or prehnite where they cut former plagioclase. For additional SEM photographs and analysis locations, see Supplementary Fig. 1

ocean floor gabbroic or ultramafic datasets (Fig. 3a). [B] in fresh basaltic rocks is so low[13] that any addition of boron should immediately increase $\delta^{11}B$ to a value in equilibrium with seawater, with further addition of boron making little difference, explaining the subhorizontal trends shown in previous data. Our in situ data show that overprinting of serpentine and saponite (AM15) by chlorite-rich assemblages and prehnite (AM16) will have led to release of boron into the fluid, only partly compensated by increased alteration of plagioclase to prehnite. At the same time, $^{11}B$ was likely fractionated into the fluid, leaving AM16 with lower $\delta^{11}B$ than AM15, as is suggested to occur in subducting slabs[8,21,42]. In the case of Hess Deep, we believe the overprinting assemblage formed during hydrothermal circulation (see below), and any boron released was most likely recycled back into the ocean via hydrothermal vents.

**Boron uptake in the lower oceanic crust and upper mantle**. In Table 1, we estimate a mean composition for Pacific lower gabbro altered in the vicinity of fault zones at <350 °C in [B], $\delta^{11}B$, $^{87}Sr/^{86}Sr$ and $\delta^{18}O$. This is based on an existing estimate of the proportion of troctolite and gabbro[32] in the lower crust, and an unpublished shipboard estimate of ~20% overprinting alteration. Our estimate assumes some undersampling of highly altered rocks during drilling. It is hard to quantify the uncertainty in our estimate beyond the standard deviation of the sample mean, but we suggest [B] and $\delta^{11}B$ are likely to be within the range of data in Table 1. We emphasise that the values from the Oman and Troodos ophiolites[39,40] are much more likely to be representative of lower gabbroic crust at fast spreading ridges away from major fault zones.

Our results show that the olivine-rich lower oceanic crust can be significantly enriched in [B], and have high values of $\delta^{11}B$, in the vicinity of fault zones, with significantly higher values of [B] than seen so far in ophiolites such as Oman. This has implications for release of boron from subducting slabs since models suggest that the temperature evolution at the Moho is very different from that of the slab surface. Models using realistic non-Newtonian rheologies in the mantle wedge[25,43] predict full dehydration of sediments and the uppermost ocean crust at the slab surface before reaching sub-arc depths for the vast majority of cases in the spectrum of hot to cold subduction, but persistence of antigorite beneath the arc at the slab Moho for the vast majority of cases. Preservation of antigorite and other hydrous phases in the upper mantle is suggested to depths where the water could be transferred to the A-phase and high pressure olivine polymorphs, and recycled into the deep mantle[44,45]. Below, we show that neither this water nor water released from the upper mantle of the slab into arc volcanics, is likely to carry a signal of oceanic boron with it, other than where transform faults or other long-lived fault zones are subducted.

The most likely way in which large parts of the slab mantle could become serpentinised is at bend faults going into subduction zones[28,29,46]. In Table 2 we follow previous authors by modelling bend fault serpentinisation as a one-pass process in which seawater is sucked downwards by low partial pressures at the reaction interface and largely absorbed by hydration of olivine[28]. We calculate the flux required to add 3% $H_2O$ to a 10 km thick layer of the upper mantle, for a uniform downward flow and also for focussed flow down faults with a spacing of 2.5 km[28] and effective widths of 100 and 30 m. The one-pass uniform flux

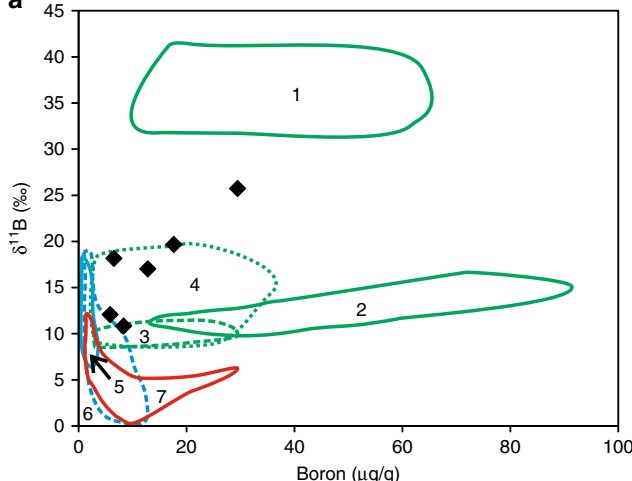

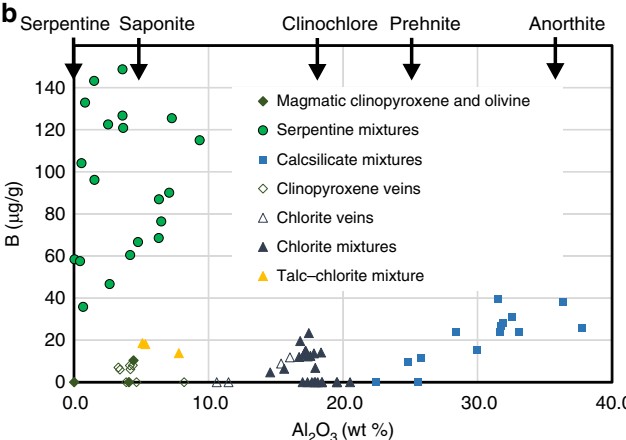

**Fig. 3** Whole rock and in situ boron analyses. **a** New whole rock data (black diamonds) compared with existing datasets: Fields 1 and 2: Atlantic core complex serpentinites[16,17]. Fields 3 and 4: Atlantic core complex talc-altered serpentinites[17,19]. Fields 5, 6 and 7: Oman ophiolite gabbros, dykes and lavas respectively[39]. **b** LA-ICPMS data for boron and $Al_2O_3$, with the $Al_2O_3$ content of endmembers indicated. Analytical RSD is 10% at 7 ppm B and 2% at 200 ppm B. However, because of uncertainties in the Si-content (to which data were normalised) of mixed analyses, the majority of the data is semi-quantitative. Calcsilicate mixtures are mainly prehnite, with minor zoisite, epidote, margarite and muscovite, as well as primary plagioclase. For analysis spots see Supplementary Figure 3, and for data see Supplementary Data 2

through the Moho required to serpentinise 30% of the olivine in a 10 km thick layer of the upper mantle (based on tomography in the Cocos plate[46]) is about $10^6$ kgm$^{-2}$. We then calculate the cumulative flux required to completely alter each layer of the crust assuming an initial [B] of 0.7 µg g$^{-1}$ and quantitative removal of B from seawater. This gives the minimum flux required to add the observed boron to each layer. This flux required for serpentinisation of the slab mantle is insufficient to add the boron observed in the lava section, let alone the gabbros. Even for the most focussed case, boron would be completely absorbed in the crustal section and only minimal boron could be added to the upper mantle. This boron could all be contained in a horizontal layer 2 m thick (Table 2), or vertical depths of about 250 m within 30 m thick fault zones. Using the Oman lower gabbro [B] makes little difference to this result.

The essence of our calculation is that the water/rock ratio required to fully serpentinise olivine is 0.13, while the ratio needed to achieve typical boron contents of serpentine is 10–50. In these circumstances hydration and addition of boron are very likely to be decoupled, as our flux calculations show. We conclude that multipass flow (in which the fluid flux is far greater than that required for the inferred degree of serpentinisation) is necessary to introduce any significant boron to the upper oceanic mantle, or indeed the lower oceanic crust if it has the [B] we observe at Hess Deep. We use "multipass" in the sense of a system in which ocean water enters the seafloor and eventually leaves it in a continuous system, in contrast to "single pass" in which all fluid is absorbed by hydration reactions or remains within the seafloor.

**Alteration in a permeable fault slot**. The only realistic way to produce multipass flow is thermal convection, which generally requires permeabilities $>5 \times 10^{-16}$ m$^2$ to operate[47,48]. Fracturing is the most likely way to achieve such permeabilities in gabbro and peridotite at Moho depths[49]. Fault zone permeability is generally dominated by the damage zone of a fault typically 100–400 m wide[50,51]. Assuming a multistrand fault zone, we present an illustrative model for fluid circulation in a permeable fault zone 2 km thick with a topographic relief typical of faulted ocean floor (Fig. 4), which is designed to model circulation and hydrothermal alteration in the recent past (0.5–1 m.y. ago) in the large normal fault zone at Hess Deep. The model is also representative of potentially longer term circulation in transform faults. We use a permeability of $3 \times 10^{-15}$ m$^2$, insufficient to support a black smoker system[52], but probably a reasonable time-averaged value in a fault zone affected by repeated slip and healing events. The basal heat flow is 220 mWm$^{-2}$, which corresponds to a plate model age of about 5 Ma[53]. Fluid moves up in the topographically highest part of the fault slot (Fig. 4c; Supplementary Fig. 4; Supplementary Movie 1), and temperatures typical of the observed alteration assemblages are seen over prolonged periods. We have then calculated flow lines within the model and integrated flux along each flow line, coupling this with our estimates of [B] in different layers of the ocean crust in Table 2. We can predict the passage of the [B] front along each flow line (Fig. 4d), and hence the time required to fully alter the fault slot. Alteration along the highest velocity flow paths penetrates to the Moho by ~250 ka, but most of the lower crust within the slot is not fully altered until ~550 ka.

Our modelling is illustrative rather than exhaustive, and we have not explored the permeability space exhaustively, nor circulation below the crustal section. Higher permeability would lead to lower fluid temperatures, and a narrower fault slot to higher temperatures, as shown in Supplementary Fig. 4. We consider it likely that undiscovered vents are present within the Hess Deep fault zone west of our study area, but possibly not at black smoker temperatures.

Our model gives a good match to the range in temperatures seen in the hydrothermal alteration assemblages at Hess Deep (Fig. 4c), and alteration could be achieved within the available time-scale. The main uncertainty is the permeability, which in a seismic fault zone will undoubtedly be highly variable both in time and space[51]. Permeability will peak during seismic events and then be reduced interseismically by mechanical closure of fractures and growth of secondary minerals, especially phyllosilicates[51]. Estimates of damage zone permeability vary widely, but recent estimates of palaeopermeability for a 4 km offset strike-slip fault in gabbro and metadiorite are significantly higher than our value[54], and a transform fault in a near-ridge environment could be the locus of more vigorous circulation and more rapid alteration than in our model.

**Table 1 Samples and data**

| I.D. | IODP sample # | Comments | $\delta^{18}O$ (VSMOW) | $^{87}Sr/^{86}Sr$ | 2sd | B µg/g | $\delta^{11}B$ | 2sd |
|---|---|---|---|---|---|---|---|---|
| AM15 | U1415P 23R1 85–90A | Troctolite, background alteration | 5.3 | 0.70346 | .00001 | 29.4 | 25.6 | 1.1 |
| AM16 | U1415P 23R1 5–90B | Troctolite, overprinting alteration | 4.1 | 0.70398 | .00004 | 12.8 | 16.9 | 0.3 |
| AM17 | U1415J 11R1 44–47 | Cataclasite overprinted by prehnite. | 5.7 | 0.70455 | .00002 | 17.6 | 19.6 | 0.5 |
| AM18 | U1415J 8R1 134–137A | Olivine gabbro; background alteration | 5.2 | 0.70265 | .00001 | 8.3 | 10.7 | 0.3 |
| AM19 | U1415J 8R1 134–137B | Olivine gabbro; overprinting alteration. Zeolite vein | 5.5 | 0.70466 | .00001 | 5.8 | 12.0 | 0.5 |
| AM20 | U1415I 2R1 25–31 | Gabbro, cataclastic, with prehnite veins, overprinting alteration | 5.5 | 0.70418 | .00001 | 6.9 | 18.1 | 0.5 |
| | Estimated average Hess Deep lower gabbro (40% AM15, 40% AM18, 10% (AM16 + AM17); 10% (AM19 +AM20) | | 5.2 | 0.70311 | | 17.2 ±10.7 | 21.7 ±1.9 | |

**Table 2 1-D integrated flux ($J_{int}$) flux calculations: mean values for lavas, dykes and upper gabbro, and Oman lower gabbro are based on Oman ophiolite[34], with dykes within the lava pile assigned to that layer. Mean values for the lower gabbro are from Table 1. Layer thicknesses based on Gillis et al. (2014)[27]**

**A: calculation of vertical water flux to Moho (based on [B])**

| Layer | Thickness (m) | Mean B (µg/g) | W/R (wgt) required | $J_{int}$ (kg/m$^2$) to alter layer |
|---|---|---|---|---|
| Lavas | 500 | 8.1 | 1.6 | $2.4 \times 10^6$ |
| Dykes | 750 | 4.5 | 0.8 | $1.9 \times 10^6$ |
| Upper gabbro | 1414 | 1.7 | 0.2 | $0.9 \times 10^6$ |
| Hess Deep lower gabbro | 2936 | 17.2 | 3.6 | $3.2 \times 10^7$ |
| Oman lower gabbro | 2936 | 1.2 | 0.1 | $0.9 \times 10^6$ |
| $J_{int}$ to Moho, Hess Deep L. gabbro (kg/m$^2$) | | | | $3.7 \times 10^7$ |
| $J_{int}$ to Moho, Oman L. gabbro (kg/m$^2$) | | | | $6.1 \times 10^6$ |

**B: Integrated flux ($J_{int}$) through Moho required to serpentinise (30%) 10 km thickness of upper mantle**

| % of Moho area through which flux occurs | H$_2$O in 10 km column of mantle (kg/m$^2$) | $J_{int}$ required through Moho (kg/m$^2$) | excess flux of H$_2$O for Hess Deep lower gabbro (kg/m$^2$) | metres of upper mantle that can be altered in B, ([B] mantle = 25 µg/g) | excess flux of H$_2$O for Oman lower gabbro (kg/m$^2$) | metres of upper mantle that can be altered in B, ([B] mantle = 25 µg/g) |
|---|---|---|---|---|---|---|
| 100 | $1.1 \times 10^6$ | $1.1 \times 10^6$ | $-3.4 \times 10^7$ | 0 | $-5.0 \times 10^6$ | 0 |
| 4 | $1.1 \times 10^6$ | $2.8 \times 10^7$ | $-9.3 \times 10^6$ | 0 | $2.2 \times 10^7$ | 0.45 |
| 0.8 | $1.1 \times 10^6$ | $1.4 \times 10^8$ | $1 \times 10^8$ | 2.0 | $1.3 \times 10^8$ | 2.64 |

We have not attempted to model convective circulation in bend faults, but discuss the likelihood of convective circulation in this setting below.

**Discussion**

Our modelling shows that water content and [B] are very likely to be decoupled in subducting slabs, and it cannot be assumed that the [B] and $\delta^{11}B$ of samples collected in anomalous settings close to the seafloor are representative of subducted hydrated mantle. Models of boron release in subduction zones have become increasingly sophisticated[6,7,21,27]. However, the biggest uncertainty in these models is the composition of the downgoing crust and mantle. For example Konrad-Schmolke and Halama[6] use [B] = 25 µg g$^{-1}$ and $\delta^{11}B = -2$ ‰ for the whole of the downgoing ocean crust, and [B] = 18 µg g$^{-1}$ and $\delta^{11}B = + 13.5$ ‰ for 30% serpentinised upper mantle. Our estimates of [B] for lithosphere affected only by bend-faulting would be the ophiolite value of 3–4 µg g$^{-1}$ for the mean ocean crust[39] and <1 µg g$^{-1}$ for the upper mantle. On the other hand transform-fault crust could be altered to the full extent of our average lower crust estimate in Table 1, and the upper mantle in transform faults could carry significant heavy boron as suggested by seafloor exposures[18].

We have shown that downward movement of seawater alone (one-pass circulation) is very unlikely to add significant boron to either the upper mantle or the olivine-rich lower oceanic crust. Data from the Oman and Troodos ophiolites[39,40] suggest that heavy B is introduced into unfaulted lower oceanic crust, but only at relatively low concentrations. Multipass circulation driven by thermal convection is required to introduce higher [B].

Bend faults generate large earthquakes, and are active over a width of ~60 km perpendicular to the trench, giving a typical period of activity of 0.5 to 1 Ma[55]. They have a spacing of 2–3 km, and offsets typically only around 300 m[28], indicating much lower slip rates than oceanic detachment faults or transform faults[28]. Damage zone widths of up to about 100 m might be expected[56]. In addition, heatflow in mature ocean crust older than 20 Ma[53] will be only 50–100 mWm$^{-2}$. Permeability in fault zones is a balance between co-seismic increase in permeability and inter-seismic reduction via vein filling and volume increase reactions[57]. Low heat flow and long recurrence times will both act to suppress hydrothermal circulation compared to near-ridge faults with higher slip rates. We cannot here rule out significant hydrothermal circulation in bend faults, but since there is an adequate one-pass model to explain the hydration of the upper mantle by bend-faulting observed in tomography[28,46], the onus is on the

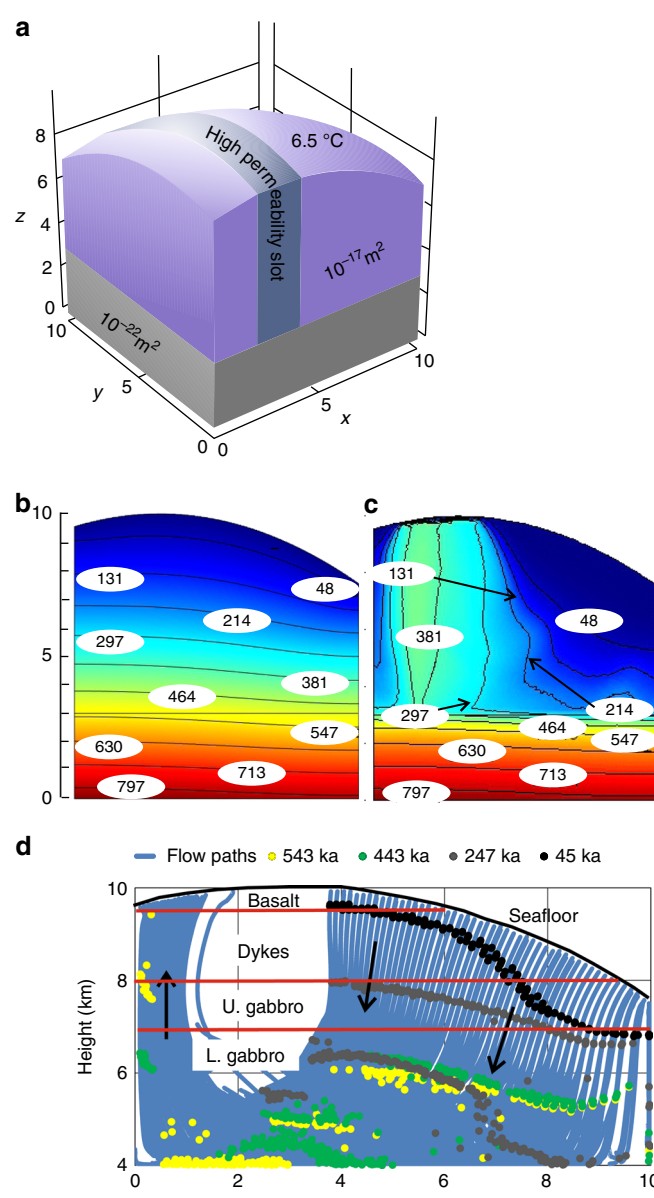

**Fig. 4** Numerical modelling of multipass hydrothermal circulation in a fault slot with submarine topography using Comsol Multiphysics™. **a** 3-D model box, with slot width 2 km, slot permeability $3 \times 10^{-15}$ m², basal heatflow 0.22 Wm$^{-2}$. All axes are in km. **b** Initial conductive thermal structure shown in a profile parallel to the fault slot. y axis is in km and the contours are in °C. **c** Thermal structure after $10^4$ years of circulation along the same profile as **b**. Vents stabilise in the shallower bathymetry with inflow in deeper bathymetry. Temperatures > 360 °C in the upflow zone are consistent with secondary clinopyroxene in the assemblage, while prehnite-chlorite and serpentine-rich assemblages could form in large volumes of the circulation system at 150–250 °C. For a snapshot of this model at $10^5$ years see Supplementary Fig. 4b. For a video of the model see Supplementary Movie 1. d Chromatographic model for the movement of the boron alteration signature using [B] for each layer as given in Table 2. Diagram is equivalent to the upper 6 km of **b**, **c**, and circulation is confined to the crust. Arrows show directions of fluid flow. Dots indicate the location of the [B] front in a particular flow path at a given time. Note that a range of flow paths through the slot are projected onto a single plane parallel to the slot, and the distance moved by the seawater boron signal depends on the integrated fluid flux along each flow path. Quantitative removal of boron from the fluid is assumed until the final [B] for each layer (see Table 2) is reached

proponents of a boron-rich slab mantle to demonstrate that hydrothermal circulation exists. We conclude that for most arcs, the observed decrease in [B] and $\delta^{11}$B with increasing depth of the slab, and the more complex signatures seen in arcs such as Kamchatka[7], are unlikely to originate in the downgoing oceanic slab mantle, however elegantly these effects may be modelled. Attention should therefore return to the forearc mantle, which is highly enriched in heavy boron[8,20,58], generally thought to be derived from the downgoing slab[23,24,59,60]. Various authors have suggested that the [B] and $\delta^{11}$B signatures in arcs may originate mainly in dragged down forearc mantle[8,61], or subduction channels[60,62,63]. Recent data from the basal peridotites of the Oman ophiolite suggests that enrichment in isotopically heavy boron is confined to a thin (<1 km) serpentinised zone above the metamorphic sole[64]. This may well be a good analogue for the basal part of the mantle wedge in a subduction zone above the altered ocean crust and sediments. It is this basal zone which is sampled by diapiric rise of serpentinite into forearc serpentine volcanos, as evidenced by blueschist clasts, derived from the downgoing slab, within the serpentine mud[24,59].

Progressive release of water and fluid-mobile elements from the downgoing slab has been modelled by various authors[6,7,21]. We do not attempt to repeat this modelling, but can address the maximum amount of boron that can be added to the forearc mantle from the downgoing slab. Transfer of boron from the slab to the forearc mantle is likely to occur by a one-pass mechanism, as modelled for bend faults in Table 2. If we use the published estimate (3.6 µg g$^{-1}$) of the average [B] of the Oman ophiolite crust[39], and if all this were removed in dehydration reactions and added to the forearc mantle, a 1 km thick zone would contain ~19 µg g$^{-1}$ of boron. If we used our values for fault-altered lower crust, the average [B] of the crust would be ~8.4 µg g$^{-1}$, and if transferred into a 1 km thick hangingwall layer as seen in the Oman ophiolite, the concentration would be ~45 µg g$^{-1}$. This is comparable to measured values in forearc serpentinites[58]. Erosion of forearc mantle[61] subjected to alteration directly by seawater at shallow levels is potentially a way of getting larger quantities of boron into the mantle wedge.

Opinions differ as to whether antigorite can be preserved to sub arc depths[25,65], but secondary olivine, clinohumite, and possibly chlorite may contain significant boron[12], and secondary olivine in high pressure rocks undergoing antigorite breakdown may contain B-rich fluid inclusions[11], which may be preserved to sub-arc depths[66]. Hence the forearc mantle may be a viable source for boron in arc volcanics even in hot subduction zones, where antigorite is predicted to break down everywhere above the slab interface before the slab reaches sub-arc depths[25].

On large offset faults such as that at Hess Deep, oceanic detachment faults, or transforms, long-lived hydrothermal circulation is more likely, and complete boron alteration of the fault-affected lower crust and significant addition of boron to serpentinised upper mantle is possible. This agrees with the observation that B/Zr in arc volcanics is higher above transform faults[9]. Significant boron can also be incorporated at high levels in slow-spread lithosphere where ultramafic rocks are commonly brought close to the seafloor by faulting. However, modern day subducting slabs are dominated by Pacific lithosphere formed at fast-spread ridges, and the contribution of slow-spread lithosphere to global subduction is small.

We conclude that incorporation of significant boron into the downgoing slab mantle will be the exception rather than the rule. Geophysical evidence for hydration of the slab mantle cannot be used to infer high contents of fluid mobile elements such as boron, which are likely to be decoupled from water addition. Models for

boron transport and release in subduction zones need to take account of the uncertainty in the input composition of the lithosphere.

## Methods

**Boron and B isotopes.** For measuring the boron abundances we used a Perkin Elmer Optima 2000 DV Inductively Coupled Plasma-Optical Emission Spectrometer (ICP-OES) hosted in the Center for Geochemical Analysis of the Geology Department of the University of South Florida. Boron was determined following fluxed fusions in a furnace at 1400 °C using $Na_2CO_3$ as flux and platinum crucibles and lids in a boron-free clean lab environment. Sample and standard preparations followed techniques described in Snyder et al.[67] and repeated analysis of internal lab standards give errors of ±5%. $\delta^{11}B$ was measured on the same powders that we used for the rest of the bulk rock elemental and isotope analysis. The sample preparation for B isotopes was conducted in the B-free labs of the CNR-IGG-Pisa, following $K_2CO_3$ fusions of ~0.5 g sample powder and purification via use of Amberlite IRA-743 boron-specific anion exchange resin, following the procedures described in Tonarini et al. (1997)[68]. The purified sample solutions were subsequently measured for B isotopes on a Neptune series multi-collector (MC)-ICP-MS at the University of Bristol following procedures described in Rae et al. (2011)[69] and references therein. Before sample analysis, the instrument was tuned and optimised for maximum $^{11}B/^{10}B$ stability[70]. Solutions were introduced using a Teflon barrel spray chamber and ~3 ml/min of ammonia gas added to improve washout. Samples were bracketed with a 50 ppb solution of NIST SRM 951 boric acid standard, which is used to correct for machine induced mass fractionation, and converts $^{11}B/^{10}B$ ratios to delta notation as permil (‰) deviation from the mean value for the SRM-951. Errors are estimated conservatively as ± 1 ‰ or better (2σ), based on replicate determinations for samples and for repeated analyses of reference materials (see Supplementary Table 1 and Supplementary Fig. 2).

**Sr isotopes.** Strontium was extracted via conventional ion-exchange chromatographic techniques in a clean laboratory facility at the University of Leeds. The total Sr blank was negligible ( < 300 pg) compared to the amount of material processed (typically ≥ 1000 ng Sr). Strontium isotope measurements were carried out at the University of Leeds on a Thermo Scientific *Triton* TIMS instrument running in static mode. The instrumental mass fractionation was corrected for by normalizing results to $^{86}Sr/^{88}Sr = 0.1194$. The repeated analysis of NIST SRM 987 standard during the course of the measurements gave an average $^{87}Sr/^{86}Sr = 0.710248 ± 6$ (2σ; n = 4) and so no further corrections of the measured Sr isotope ratios were necessary. We also measured the Sr isotope ratios of BHVO- 1 (Hawaii basalt) and JB-3 (Japanese arc basalt) standards which were prepared and analysed together with the Hess Deep samples. These standards gave values of $^{87}Sr/^{86}Sr = 0.703488 ± 4$ and $^{87}Sr/^{86}Sr = 0.703400 ± 6$, respectively.

**Mineral chemistry.** Mineral major element analyses were performed at the University of Leeds using a Jeol JXL 8230 Superprobe. An accelerating voltage of 15 kV was used. Beam currents were typically 15 nA and a beam size of 10 μm was used throughout the study. Data are presented in Supplementary Data 1, and identified on Fig. 2 and Supplementary Figure 1. Most data are mixed phyllosilicates, and analyses with totals < 80% were excluded.

**Laser ablation - inductively coupled plasma mass spectrometry (LA-ICPMS).** LA-ICPMS analysis of polished sections was carried out at the School of Earth and Environment, University of Leeds[71,72]. Samples were ablated with a 193 nm Geolas Q Plus excimer laser (ArF, 193 nm, Microlas, Göttingen, Germany) with a 50 μm spot diameter. The ablated material was analysed in an Agilent 7500c Quadrupole ICP-MS equipped with an Octopole Reaction System where $H_2$ was added to remove interferences on the main isotopes of Ca and Fe. The recorded isotopes were $^7Li$, $^{11}B$, $^{23}Na$, $^{24}Mg$, $^{27}Al$, $^{29}Si$, $^{39}K$, $^{40}Ca$, $^{56}Fe$, $^{88}Sr$ and $^{137}Ba$ with 10 ms dwell times for all except B (50 ms) and Li (20 ms). NIST silicate glass standard SRM 610 and SRM1412 were used as calibration standards to convert the element ion signals to wt/wt ratios relative to the internal standard element. NIST SRM 612 was used as a check of the analysed concentrations. Signal integration was performed with the Matlab ®-based SILLS program[73]. The accuracy of boron concentrations was checked by analysis of serpentine standards 'Geiss' and '21826', amphibole 21805 previously analysed by SIMS[74] and Lipari obsidian standard B-6[75].

The composition of the analysed spots was calculated by normalisation to the Si intensity. The fine grain size of the alteration products led to most laser spots sampling mixtures rather than single phases. Consequently the appropriate $SiO_2$ concentration for use in the normalisation was estimated as follows: For the calcium-rich spots the analyses were interpreted as prehnite-chlorite mixtures based on Mg content. For phyllosilicate mixtures, the analyses were interpreted as mixtures of serpentine, chlorite and saponite or talc based on relative proportions of Si, Al, Mg + Fe and Na. For clinopyroxene and olivine the ideal formulae were used.

The data are presented in Supplementary Data 2, and spot locations shown in Supplementary Figure 3. Uncertainty in the Si concentration in mixtures of phases means that the LA-ICPMS data should be viewed as semi-quantitative.

**Numerical modelling.** Full details of the modelling methodology are given in Titarenko and McCaig[47], and only a brief summary is given here. The problem was calculated on a quadro core workstation with COMSOL multiphysics™ software using the Heat Transfer and Subsurface Flow Modules [COMSOL Multiphysics, 2013]. To calculate fluid flow pathways and boron alteration we used an additional program written in C + + . We used pure water properties calculated according to the international steam tables IAPWS-IF97[76]. To avoid the issue of phase separation we adopted simplified water properties, by calculating pure water properties along an isobar at 50 MPa (corresponding to the approximate pressure in the centre of the model domain). Over the temperature range of 0–300 °C experienced by fluids at most points in the model, the effects of pressure on water density and viscosity are small compared to those of temperature[76], and pure water properties show very similar temperature dependence to seawater[77,78]. Therefore the fluid flow system can be described by a set of equations for one phase[47].

Rock properties $C_{PP}$ $(T)$, $k_p$ $(T)$ and $\kappa$ $(T)$ are temperature dependent functions[79,80]. Graphs of all temperature functions can be found in Titarenko and McCaig (2015)[47]

Temperature-dependent permeability functions designed to simulate ductile closure of cracks at high temperature were used[81,82]. The permeability falls to $10^{-22}$ $m^2$ at temperatures of 800 °C from a permeability value below 600 °C, which depends on the domain within the model.

We used a constant heat flow lower boundary condition of 0.22 $W/m^2$. For low temperature off-axis circulation this is much more realistic than the constant temperature boundary condition generally used in magmatic hydrothermal models. For the top boundary condition we used a constant temperature of 6.5 °C, as discussed in Titarenko and McCaig (2015)[47]. We allow fluid to move through the upper boundary only, with the bottom and sides impermeable.

Initial temperature has been calculated by solving the purely conductive problem over the whole domain. Setting the basal heat flux to 0.22 $W/m^2$ and temperature on the seafloor to 6.5 °C, we get the initial temperature distribution shown in Fig. 4b.

The problem was solved on a 3-D domain with a topographic upper surface (see Fig. 4a). For discretization, a Delaunay triangulation algorithm was used. We applied a nonuniform grid, refining the mesh size for the domains with more vigorous convection.

**Code availability.** Numerical modelling was conducted using the proprietary code Comsol Multiphysics™. Source code cannot be made available, but the model could be reproduced from the information given if a Comsol licence were purchased. A C++ program was written to extract data from the model and calculate the position of boron fronts (This could also be achieved using the Matlab interface available in Comsol). The C++ code cannot be used without Comsol outputs, but is available from the authors on request. Final graph plotting for Fig. 4d was done using Microsoft Excel. This file is available from the authors on request.

## Data availability

All electron probe, La-ICPMS and geochemical data used in this submission is presented either in the main text or in the supplementary material.

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

## Acknowledgements

We thank Jeff Ryan and Zac Atlas (both at University of South Florida) for obtaining the B concentrations, James Rae and Tim Elliott for the analytical expertise in the analysis of the [11]B/[10]B ratios, Richard Walshaw and Duncan Hedges for assistance with SEM and Electron Microprobe work, and Patrick Sugden for assisting with the chemistry on boron isotope samples. This work was supported by NERC grant NE/K011030/1 to A.M.C.

## Author contributions

A.M.C. described and collected samples, collected SEM and microprobe data, and performed the calculations in Table 2. S.S.T. undertook the numerical modelling. I.P.S. and S.A. conducted boron isotope analysis including establishing a new MC-ICPMS methodology at Bristol University. R.A.C. conducted strontium isotope analyses and A.B. oxygen isotope analyses, and assisted D.B. with the LA-ICPMS data collection and reduction.

## Additional information

**Competing interests:** The authors declare no competing interests.

