## [Peer Review File · Nature Communications]

Reviewer #1 (Remarks to the Author):

General overview.

This paper presents interesting new results for samples of deep oceanic crust sampled in situ by ODP drilling. Opportunities to directly sample this type of reservoir are rare and inherently publication worthy. That the results for boron geochemistry differ from observations on analog outcrops makes them particularly interesting.

The authors make the case that transport of seawater boron into the deep oceanic crust is presumably via hydrothermal circulation while the rocks are near an active oceanic spreading center. Modeling of fluid circulation implies that most seawater boron inventory is consumed in this process, and that relatively little boron is likely to be transported to greater (i.e., mantle) depths. In other words, most boron in the oceanic lithosphere is concentrated near the surface. It is implied that, upon subduction of such lithosphere, the primary source of 'excess boron' observed in volcanic arc magmas is likely derived from this thin selvage on the subducting slabs.

This conclusion contrasts with many papers that attribute the source of arc boron to deeper serpentinized domains in subducting slabs. In essence, the authors believe that the mantle portion of slabs (at least as derived from their origins at mid-ocean ridges) essentially is too poor in boron to be a significant source for arc magma boron. If true, this could simplify understanding of arc magmatism.

However, toward the end of the paper they bring up the issue regarding the possible tectonic erosion and subduction of serpentinized rocks from the forearc mantle wedge. Such domains are likely to be metasomatized by fluids driven off of shallow segments of subducting slabs, and could in my opinion provide a work-around of their hypothesis. That is, serpentinization and boron-enrichment of mantle rocks could be a multi-stage phenomenon.

I think that the fundamental data and modeling are definitely worth publication, but I would recommend that the authors also quantitatively address the latter scenario to make their paper more comprehensive.

I have annotated the Word version of the paper with a variety of comments. In places, I think the writing can be improved, and typos fixed (including several of the references).

I also suggest adding a few additional references, notably:

Ryan & Langmuir (1993) GCA, B geochemistry of arc and other lavas

Tonarini et al. (2011) EPSL, S. Sandwich arc and tectonic erosion of forearc mantle

Leeman & Sisson (1996) Rev. Mineralogy review paper, that addresses the subduction inventory for boron

I have not really assessed the numerical modeling as that is beyond my expertise, and hopefully an expert is looking into that.

Bill Leeman

Reviewer #2 (Remarks to the Author):

The manuscript by McCaig et al. deals with the alteration of deep oceanic gabbro from the Pacific plate, now exposed along the Hess Deep fault scarp (Galapagos triple junction). The paper is important because it documents for the first time that this ocean crust (involved in present-day subduction) uptakes significant amounts of B and of the heavy B isotope (^{11}B) via exchange with seawater-derived fluids. Being these rocks exposed along a normal fault system, it is envisaged by the authors that the fluid-gabbro interaction occurs during deep seawater infiltration along the fault, comparably to other fault systems like transform and bend faults. The latter, located at outer rises near subduction-zones trenches, are thought to facilitate seawater infiltration into the lithospheric mantle, allowing peridotite serpentinization water recycling into subduction zones. Main result of this study is that seawater-gabbro interaction consumes all H_2O , B and ^{11}B , which become stored in secondary alteration minerals. This way, as suggested by modelling, in a short-lived fault recording a one-pass circulation system, all H_2O and B are removed from fluids by interaction with crustal gabbros and cannot be added to the underlying mantle. Main implication is that the oceanic Pacific mantle becomes rarely hydrated near short-lived faults and cannot be a water and boron source for subduction and arc magmatism. Another H_2O and B source must be found elsewhere, and the authors individuate the hydrated fore-arc mantle as carrier of water and boron released to arc magmas.

The submitted study is of general interest, the data are of good quality and the scenario proposed is possible. Overall, the argument is suitable to Nature Communications. The paper can thus be considered for publication, provided that key aspects like the rock petrography and microstructure,

and the presentation of the model are sufficiently strengthened and clarified. This requires major revision of the present manuscript. Hereafter I list my general comments.

1. The six gabbro samples analysed are a few for such a global picture, and the sample set does not include the mantle rocks exposed along the fault scarp (page 4, line 78). I am aware that in IODP cruises direct sampling and sample availability are restricted. However, since the proposal is that crustal gabbro takes up water and B and mantle doesn't, it could be relevant having analyses and indications from mantle rocks.

2. I do not fully understand the point that authors want to make at page 3 (lines 55-60): "the input boron concentration and $\delta^{11}\text{B}$ used in these models for subducted slab serpentinites are based almost entirely on samples collected from the seafloor in the Atlantic. These are exposed due to large-displacement faulting in fracture zones or oceanic core complexes, and cannot be considered typical of the upper mantle in relatively intact oceanic mantle entering subduction zones". What do they mean with intact oceanic mantle entering subduction zones? The Atlantic ridge includes large domains where magmatism is scarce and mantle outcrops at the ocean floor, and core complexes of deep gabbro intrusions and host mantle are exhumed along detachment faults. The Atlantic mantle is thus tectonically troubled, exposed near surface, serpentinitized and B rich and, I agree, it can be different from the presently subducted Pacific one. Anyway, it should be stressed that an Atlantic-type ocean was subducted in the past being therefore representative of fossil subduction settings. However, even today, there is no intact mantle entering subduction zones: at outer rises, the bend faults dissect the mantle and yet none knows what the extent of mantle serpentinitization around these faults is, if hydrothermal water circulation occurs there and what heat budgets are. The authors should clarify these points and what they mean with intact mantle entering subduction zones.

3. Pages 2-3, lines 62-63. The authors state that the studied fault environment is "analogous to fault-related alteration in bend faults or transform faults". This is not fully ascertained for bend faults (see point 2 above). However, at page 11-12 (lines 259-262) it is stated that: "On large offset faults such as that at Hess Deep, oceanic detachment faults, or transforms, long-lived hydrothermal circulation is more likely, and complete boron alteration of the fault-affected lower crust and significant addition of boron to serpentinitized upper mantle is possible". This is misleading because the model proposed in this paper for Hess Deep (Fig. 4) regards a short-lived fault, with single pass fluid flow that doesn't lead to mantle serpentinitization. Therefore, the authors should clearly state which kind of fault settings are actually comparable with the Hess Deep, and must clearly point out the difference/similarity with the bend faults. As it is, the text leaves the reader a bit puzzled.

4. Page 4-5, lines 86-102. The petrographic and textural description of rocks is unclear, and needs more detail and proper wording. It is unclear what the authors mean with background and green alteration. Is the serpentine + talc background alteration overgrown by subsequent (green?) mineral

assemblages? I think authors must clarify what they mean with background and green alteration, and which microstructural relationships exist between background and green alteration minerals. These rocks seem to me to record two superposed alteration stages, an earlier one (at ridge?) and a second one along cataclastic zones and cracks due to faulting. If so, this must be clarified in the text using appropriate figures. Figure 2 is difficult to read, has too many unexplained areas, it doesn't clarify what clinopyroxene means in terms of petrologic evolution, and doesn't clearly show any superposition between different mineral alteration assemblages. If two distinct alteration events occur in the Hess Deep gabbro, how do they relate to the fault model presented in Figure 4? All alteration (background and green) is modelled in a single pass fluid flow in a fault environment, but my feeling is that faulting just brings to a part (not all) of alteration recorded by the analysed rocks. Moreover, it is difficult to judge if part of the initial (background) alteration can be achieved without faulting and how deep this alteration can penetrate. Finally, can one exclude that the whole alteration process (background and green) occurred during seafloor exposure of these rocks and has nothing to see with deep faulting environments? How do these uncertainties affect the model presented in Figure 4?

5. The bulk rock analyses presented and discussed at page 5-6 are very good quality bulk rock analyses. However, if the alteration is accomplished in two stages instead of one, single pass down-flow of fluid in a fault, it could be good to have in-situ mineral analyses showing what is the budget achieved during each alteration stage (background and green).

6. Page 9, Fig. 4. Modelling is set to represent a transform fault or the normal fault at Hess Deep, and it can explain what is going on there in terms of water and boron partitioning between rocks and fluid, provided that background alteration stage 1 indeed took place during faulting. This model is also used for envisaging behaviour of bend faults, although these are not well known and one cannot exclude multi-pass circulation there. Considering Figure 4, its presentation needs improvement by clarifying: (1) what the x and y axes do represent in Figure 4, what is the thickness of the permeability zone, what the values reported in the figure mean; (2) what is reported as y axis in Fig. 2b and what the numbers inside the white ellipses do represent; (3) where the mantle is expected to be in Fig. 4d and what the distance reported in the x axis means (distance from what?).

7. This paper suggests that water and B are consumed by the oceanic crust from fluids fluxing downwards along a fault formed in near ridge environments. If so, the temperature in such environments is higher compared to bend faults, which form in colder environments. It is possible that reaction kinetics at bend faults is sluggish compared to other settings and that seawater there reaches the mantle without losing its B and ¹¹B load.

Minor comments

Abstract. Line 15. Boron is a major proxy for seawater release. Other tracers like halogens are equally important. At the end of the abstract it could be good to add a sentence telling that the serpentinized fore-arc mantle is a suitable domain for water and B transfer to arcs.

Line 49. Not only fore-arc mantle but also serpentinite embedded at the subduction interface can uptake slab fluids and boron.

Line 75-76. Unclear sentence.

Reply to reviewers NCOMMS-17-28940

Reviewer #1 (Remarks to the Author):

General overview.

This paper presents interesting new results for samples of deep oceanic crust sampled in situ by ODP drilling. Opportunities to directly sample this type of reservoir are rare and inherently publication worthy. That the results for boron geochemistry differ from observations on analog outcrops makes them particularly interesting.

The authors make the case that transport of seawater boron into the deep oceanic crust is presumably via hydrothermal circulation while the rocks are near an active oceanic spreading center. Modeling of fluid circulation implies that most seawater boron inventory is consumed in this process, and that relatively little boron is likely to be transported to greater (i.e., mantle) depths. In other words, most boron in the oceanic lithosphere is concentrated near the surface. It is implied that, upon subduction of such lithosphere, the primary source of 'excess boron' observed in volcanic arc magmas is likely derived from this thin selvage on the subducting slabs.

This conclusion contrasts with many papers that attribute the source of arc boron to deeper serpentinized domains in subducting slabs. In essence, the authors believe that the mantle portion of slabs (at least as derived from their origins at mid-ocean ridges) essentially is too poor in boron to be a significant source for arc magma boron. If true, this could simplify understanding of arc magmatism.

However, toward the end of the paper they bring up the issue regarding the possible tectonic erosion and subduction of serpentinized rocks from the forearc mantle wedge. Such domains are likely to be metasomatized by fluids driven off of shallow segments of subducting slabs, and could in my opinion provide a work-around of their hypothesis. That is, serpentinization and boron-enrichment of mantle rocks could be a multi-stage phenomenon.

This suggestion was in the original submission, but we have enlarged upon it, with additional data, for example recent data on the basal serpentinised peridotite of the Oman ophiolite

I think that the fundamental data and modeling are definitely worth publication, but I would recommend that the authors also quantitatively address the latter scenario to make their paper more comprehensive.

We have addressed the question of the maximum amount of boron that could be added from the downgoing crust to the overlying mantle or melange zone. It is not really possible in this paper to address quantitatively the question of forearc erosion, where the composition of the material being eroded is uncertain

I have annotated the Word version of the paper with a variety of comments. In places, I think the writing can be improved, and typos fixed (including several of the references).

I also suggest adding a few additional references, notably:

Ryan & Langmuir (1993) GCA, B geochemistry of arc and other lavas

Tonarini et al. (2011) EPSL, S. Sandwich arc and tectonic erosion of forearc mantle
Leeman & Sisson (1996) Rev. Mineralogy review paper, that addresses the subduction inventory for boron

The majority of the changes identified on the manuscript have been made, and the above references added

I have not really assessed the numerical modeling as that is beyond my expertise, and hopefully an expert is looking into that.

Bill Leeman

Reviewer #2 (Remarks to the Author):

The manuscript by McCaig et al. deals with the alteration of deep oceanic gabbro from the Pacific plate, now exposed along the Hess Deep fault scarp (Galapagos triple junction). The paper is important because it documents for the first time that this ocean crust (involved in present-day subduction) uptakes significant amounts of B and of the heavy B isotope (^{11}B) via exchange with seawater-derived fluids. Being these rocks exposed along a normal fault system, it is envisaged by the authors that the fluid-gabbro interaction occurs during deep seawater infiltration along the fault, comparably to other fault systems like transform and bend faults. The latter, located at outer rises near subduction-zones trenches, are thought to facilitate seawater infiltration into the lithospheric mantle, allowing peridotite serpentinization water recycling into subduction zones. Main result of this study is that seawater-gabbro interaction consumes all H_2O , B and ^{11}B , which become stored in secondary alteration minerals. This way, as suggested by modelling, in a short-lived fault recording a one-pass circulation system, all H_2O and B are removed from fluids by interaction with crustal gabbros and cannot be added to the underlying mantle. Main implication is that the oceanic Pacific mantle becomes rarely hydrated near short-lived faults and cannot be a water and boron source for subduction and arc magmatism. Another H_2O and B source must be found elsewhere, and the authors individuate the hydrated fore-arc mantle as carrier of water and boron released to arc magmas.

There appears to be a misunderstanding of part of our argument, because we say that H_2O can be added without B and ^{11}B . We have tried to make this decoupling clearer in various places

The submitted study is of general interest, the data are of good quality and the scenario proposed is possible. Overall, the argument is suitable to Nature Communications. The paper can thus be considered for publication, provided that key aspects like the rock petrography and microstructure, and the presentation of the model are sufficiently strengthened and clarified. This requires major revision of the present manuscript. Hereafter I list my general comments.

1. The six gabbro samples analysed are a few for such a global picture, and the sample set does not include the mantle rocks exposed along the fault scarp (page 4, line 78). I am aware that in IODP cruises direct sampling and sample availability are restricted. However, since the proposal is that crustal gabbro takes up water and B and mantle doesn't, it could be relevant having analyses and indications from mantle rocks.

There are some [B] data available from Hess Deep serpentinised mantle rocks collected by Kodolanyi et al 2012 which we have referred to. These rocks show enrichment in [B] similar to the lower crustal gabbros we sampled, but since this is another near-seafloor dataset, it only has relevance to fault-affected enrichment, as with the Atlantic data. There is no data available on $\delta^{11}\text{B}$. It is beyond the scope of this revision to resample IODP cores and create more data on this aspect, but we have added a $\delta^{11}\text{B}$ value for one of our six samples which did not run first time.

2. I do not fully understand the point that authors want to make at page 3 (lines 55-60): "the input boron concentration and $\delta^{11}\text{B}$ used in these models for subducted slab serpentinites are based almost entirely on samples collected from the seafloor in the Atlantic. These are exposed due to large-displacement faulting in fracture zones or oceanic core complexes, and cannot be considered typical of the upper mantle in relatively intact oceanic mantle entering subduction zones". What do they mean with intact oceanic mantle entering subduction zones? The Atlantic ridge includes large domains where magmatism is scarce and mantle outcrops at the ocean floor, and core complexes of deep gabbro intrusions and host mantle are exhumed along detachment faults. The Atlantic mantle is thus tectonically troubled, exposed near surface, serpentinized and B rich and, I agree, it can be different from the presently subducted Pacific one. Anyway, it should be stressed that an Atlantic-type ocean was subducted in the past being therefore representative of fossil subduction settings. However, even today, there is no intact mantle entering subduction zones: at outer rises, the bend faults dissect the mantle and yet none knows what the extent of mantle serpentinization around these faults is, if hydrothermal water circulation occurs there and what heat budgets are. The authors should clarify these points and what they mean with intact mantle entering subduction zones.

We have tried to clarify this in the Discussion. The overwhelming majority of lithosphere currently being subducted was generated at fast-spreading ridges in the Pacific, not slow spreading ridges (this is a function of the need to balance creation and destruction of lithosphere, and so was almost certainly true in the past. However, since we are primarily dealing with chemical variations and processes in modern arcs, we do not attempt to make this point in the paper. We agree that bend faults are a locus of serpentinisation, but a central point of this paper is that these serpentinites will only exceptionally be enriched in boron, because large scale hydrothermal circulation on bend faults is unlikely and has not been observed

3. Pages 2-3, lines 62-63. The authors state that the studied fault environment is "analogous to fault-related alteration in bend faults or transform faults". This is not fully ascertained for bend faults (see point 2 above). However, at page 11-12 (lines 259-262) it is stated that: " On large offset faults such as that at Hess Deep, oceanic detachment faults, or transforms, long-lived hydrothermal circulation is more likely, and complete boron alteration of the fault-affected lower crust and significant addition of boron to serpentinized upper mantle is possible". This is misleading because the model proposed in this paper for Hess Deep (Fig. 4) regards a short-lived fault, with single pass fluid flow that doesn't lead to mantle serpentinization. Therefore, the authors should clearly state which kind of fault settings are actually comparable with the Hess Deep, and must clearly point out the difference/similarity with the bend faults. As it is, the text leaves the reader a bit puzzled.

Hopefully we have clarified our point here. Hess Deep is a large offset fault compared to bend faults, and is located in a regime near-ridge of high heat flow. We compare models of single pass and multipass fluid flow to show the difference between them in terms of geochemistry. Fig. 4 is the multipass model, not the single pass model which is described in Table 2. In short, Fig. 4 is our

proposed model for the intense alteration seen in lower crustal rocks at Hess Deep, and we speculate that similar alteration will occur on long-lived transform faults and detachment faults. The one-pass model has been suggested by Ranero et al for bend faults and we test whether this published model can enrich the slab mantle in boron

4. Page 4-5, lines 86-102. The petrographic and textural description of rocks is unclear, and needs more detail and proper wording. It is unclear what the authors mean with background and green alteration. Is the serpentine + talc background alteration overgrown by subsequent (green?) mineral assemblages? I think authors must clarify what they mean with background and green alteration, and which microstructural relationships exist between background and green alteration minerals.

We have clarified the relationship between the two types of alteration as far as possible by using the words background and overprinting more consistently. We have also changed the electron probe data presentation in Fig 2d from a binary to a ternary plot, which shows the relationship between phyllosilicate compositions in the background and overprinting alteration more clearly.

These rocks seem to me to record two superposed alteration stages, an earlier one (at ridge?) and a second one along cataclastic zones and cracks due to faulting. If so, this must be clarified in the text using appropriate figures. Figure 2 is difficult to read, has too many unexplained areas, it doesn't clarify what clinopyroxene means in terms of petrologic evolution, and doesn't clearly show any superposition between different mineral alteration assemblages.

The two alteration types show transitional rather than sharp changes at the scale of a thin section, but the overprinting alteration occurs in vein halos and in more faulted samples, as shown in Fig.1. Fig. 2 does need to be looked at together with supplementary figure 1, which shows enlargements of the other areas. The big difference petrographically is the disappearance of the serpentine mesh texture and the serpentine-saponite assemblage, and any primary minerals other than clinopyroxene in the overprinting assemblage.

As is often the case with alteration halos, it is hard to distinguish between events widely separated in time, and progressive alteration. All the rocks sampled at Hess Deep come from within a large fault zone, and our preferred model is that the background alteration occurred mostly in this environment, rather than at the EPR before faulting started. We clearly state that the Oman ophiolite data published by Yamaoka et al is a better model for the boron content and isotopic composition of unfaulted ocean crust. No in situ data from fast spread crust can be collected other than in anomalous faulted environments until we have a Mohole through an unfaulted section, which is of course proposed.. It is a great weakness of models of fluid-mobile element release in subduction zones that they rely on "representative" data collected in highly unrepresentative places. This is a central point of our paper, and we fully accept that our data is only representative of "fault-altered" lower crust. We do have more detail from other samples of the alteration process and conditions, but that is a long paper in itself and will be published elsewhere.

If two distinct alteration events occur in the Hess Deep gabbro, how do they relate to the fault model presented in Figure 4? All alteration (background and green) is modelled in a single pass fluid flow in a fault environment, but my feeling is that faulting just brings to a part (not all) of alteration recorded by the analysed rocks. Moreover, it is difficult to judge if part of the initial (background) alteration can be achieved without faulting and how deep this alteration can penetrate. Finally, can one exclude that the whole alteration process (background and green) occurred during seafloor

exposure of these rocks and has nothing to see with deep faulting environments? How do these uncertainties affect the model presented in Figure 4?

The reviewer highlights various uncertainties that are present in virtually all alteration environments. As stated above, since all the samples have been collected from a fault scarp, it will never be certain how much of the background alteration may have formed prior to that faulting, and how much during it. However we can rule out dominant very near seafloor low temperature alteration, since the oxygen isotope data is not consistent with that (which would give positive deviations from the original crustal value of around +5.5 per mil), and the overprinting secondary clinopyroxene is consistent with temperatures above 360 C (as shown in a diagram in the IODP cruise report, which is cited). Nevertheless Fig 4c shows that during active hydrothermal circulation, temperatures can be very variable within 1 km of the seafloor. We already provide more detailed petrographic data than most previous studies of boron in altered oceanic or ophiolitic settings, and have added in-situ boron data from the same samples as requested below

5. The bulk rock analyses presented and discussed at page 5-6 are very good quality bulk rock analyses. However, if the alteration is accomplished in two stages instead of one, single pass down-flow of fluid in a fault, it could be good to have in-situ mineral analyses showing what is the budget achieved during each alteration stage (background and green).

We have provided a significant dataset on new LA-ICPMS data on boron which supports our model that boron is lost and fractionated during the overprinting alteration. This still is entirely possible during a single fault-related alteration, but at variable temperature as indicated in Fig. 4

6. Page 9, Fig. 4. Modelling is set to represent a transform fault or the normal fault at Hess Deep, and it can explain what is going on there in terms of water and boron partitioning between rocks and fluid, provided that background alteration stage 1 indeed took place during faulting. This model is also used for envisaging behaviour of bend faults, although these are not well known and one cannot exclude multi-pass circulation there. Considering Figure 4, its presentation needs improvement by clarifying: (1) what the x and y axes do represent in Figure 4, what is the thickness of the permeability zone, what the values reported in the figure mean; (2) what is reported as y axis in Fig. 2b and what the numbers inside the white ellipses do represent; (3) where the mantle is expected to be in Fig. 4d and what the distance reported in the x axis means (distance from what?).

The figure caption has been clarified to answer the above questions. Distance along the x axis is distance in the model, which is generic, not Hess Deep in particular.

7. This paper suggests that water and B are consumed by the oceanic crust from fluids fluxing downwards along a fault formed in near ridge environments. If so, the temperature in such environments is higher compared to bend faults, which form in colder environments. It is possible that reaction kinetics at bend faults is sluggish compared to other settings and that seawater there reaches the mantle without losing its B and 11B load.

This is possible, but the total amount of boron that can be added to the lithosphere in one-pass circulation is still much smaller than the estimates of crustal and mantle boron contents used by Konrad-Smolke and others

Minor comments

Abstract. Line 15. Boron is a major proxy for seawater release. Other tracers like halogens are

equally important. At the end of the abstract it could be good to add a sentence telling that the serpentinized fore-arc mantle is a suitable domain for water and B transfer to arcs.

At the moment the abstract is at the limit of permitted length according to Nature Communications guidelines. We agree it could benefit from an extra couple of sentences, if allowed

Line 49. Not only fore-arc mantle but also serpentinite embedded at the subduction interface can uptake slab fluids and boron.

We have mentioned subduction melanges, which is presumably what this is alluding to

Line 75-76. Unclear sentence.

Reviewer #1 (Remarks to the Author):

Overall, I think this paper is acceptable for publication. The authors have addressed reviews adequately in my opinion. The text has been improved and I feel laced with sufficient caveats that their main point is credible – namely that in most instances (where hydrothermal circulation is limited), it is unlikely that there will be substantive addition of B to the upper mantle portions of most subducting slabs. Rather, it would be likely that B enrichments may be concentrated in the oceanic crustal section of such slabs. They allow that additional sources of ‘excess’ B observed in many arc magma suites could include slab domains that have been strongly influenced by penetration of hydrothermal fluids (e.g., fracture zones), or tectonically eroded frontal arc materials that have partially equilibrated with seawater.

Note the following comments:

p. 6, l. 20 typo ‘born’ -> boron

l. 22 Is your ‘global average’ based just on the few samples you analyzed? Maybe you could define what this really means? The real question is ‘what does such an average really represent?’

p. 7 l. 12 given this caveat, what are estimated uncertainties on the LA-ICP data? Is this trivial or significant? Perhaps indicate approximate error bars on Fig. 3b.

p. 10 l. 10 typo ‘ration’ -> ratio

p. 14 l. 8 typo ‘1km’ -> 1 km

l. 10 typo ‘seem’ -> seen

Re: methods – I am curious as to why the B isotopic measurements were not made in Pisa (since the B was extracted there)? I would expect that the analytical error by TIMS method would be smaller than that obtained at Bristol.

Re: modeling – Fig. 4 seems to represent multipass hydrothermal circulation (see caption), wherein presumably there is repeated (continuous?) infiltration of fluid (seawater?). Panel [d] show depths

of B-enrichment fronts as a function of time (up to ca. 0.5 Ma). According to this model, it would appear that B-enrichment is largely confined to the oceanic crust.

Accordingly, for your interpretation, it appears that you consider hydrothermal circulation to be generally confined to such short duration. I may have missed it, but perhaps you should explicitly state this – maybe in the caption.

Also, for comparison, it would be instructive to show comparable model outcomes for a long-lived fracture zone like Amlia (Aleutians), and for forearc ultramafic debris as proposed for S. Sandwich subduction zone. I expect this might be feasible without too much effort, and it would certainly help readers visualize differences associated with these 'end member' scenarios.

William Leeman

Reviewer #2 (Remarks to the Author):

This second version of the paper "No significant boron in the hydrated mantle of subducting slabs" has improved in clarity. The authors properly addressed the reviewers' comments, providing more clear arguments supporting the conclusions. The main message is now clear, provocative and suitable for the Journal. Not being an expert, I can't fully judge the modelling part of the work.

However, some queries I raised to the first version of this manuscript still need clarification. The paper states that the serpentinized slab mantle is not sufficiently B and ¹¹B-rich to affect B release to arcs. An alternative B source for subduction fluids must be found, and the authors propose the subduction-erosion of serpentinized forearc mantle. I do agree, though I think this holds true for present day subduction of the Pacific fast-spreading ocean, whose thick crustal gabbroic layer consumes B from infiltrating seawater. This conclusion may not apply to past subduction of slow-spreading oceans with different lithospheric structure. Therefore, I agree that present day slab serpentinites are B-free, but I would be cautious for past subduction systems. As conveyed in the manuscript, this conclusion applies to all subducting slabs, and I suggest smoothing this aspect in the title (e.g. No significant boron in the hydrated mantle of present-day subducting slabs) and in the text.

Specific comments

1. Abstract. Last sentence "We conclude that only in unusual circumstances can the upper mantle of subducting slabs be a source of boron enrichment in arc volcanics, or deep mantle recycling". As explained above, the statement that serpentinized slab mantle is B-free is fine for the present case-study, and it must be mentioned this may not apply to past subduction systems, as suggested by many ophiolitic slab-serpentinites. This sentence may change to: We conclude that only in unusual circumstances can the upper mantle of fast-spreading oceans undergoing present-day subduction be a source of boron enrichment in arc volcanics, or deep mantle recycling.

2. Background section, page 4. Sentence: "typical of the upper mantle in relatively intact oceanic mantle entering subduction zones". I apologize, but it still is unclear to me what this intact mantle is. I think it should be clearly stated that this mantle occurs in present-day fast spreading plates.

3. Mineral Data section, page 8. The data suggest that part of the B gained during initial background alteration is back-released to fluids during formation of the green alteration, and that ^{11}B fractionates into the fluid. Can this new ^{11}B -bearing fluid affect serpentinization of the mantle rocks underneath?

4. Mineral Data section, page 9. Sentence " Our results show that the olivine-rich lower oceanic crust can be significantly enriched in [B], and have high values of $\delta^{11}\text{B}$, in the vicinity of fault zones, with much higher values of [B] than seen so far in ophiolites". Figure 3a shows that the B and ^{11}B of the Oman gabbros are comparable to those presented in this paper, with the only exception of two enriched samples. This should be better clarified.

Figure 1. The labels AM15, AM16, overprint, cpx vein are not clearly visible. In the caption it should be explained that the sample from Fig. 1a is divided into part AM15, showing preferential development of the background alteration and into part AM16 characterized by preferential development of the green alteration. Correct (cpyx) into (cpx)

Reviewer #3 (Remarks to the Author):

Review of McCaig et al. (2018), "No significant boron in the hydrated mantle of subducting slabs"

Review by Andrew T. Fisher, 8/26/18

This is a study of rock petrology/alteration and seafloor hydrogeology, examining if and how [B] and B isotopic values in crust and mantle rocks might be influenced by fluid circulation. The key petrographic comparison is between rock samples collected at multiple field sites on land and on/below the seafloor, whereas the key hydrogeologic comparison is between single-pass and multiple-pass hydrothermal circulation.

I was not a reviewer of the original submission, but was asked to comment on the revision wrt the hydrogeologic modeling. I'll note up front that I share the author's skepticism about single-pass circulation being a ubiquitous and pervasive process along flexural faults at subduction zones. My skepticism extends to multipass circulation in flexural faults. My biggest concern is with the lack of a consistent thermal signal that should be associated with this process, if it were to occur – it would generate a readily detectable seafloor heat flux anomaly. The lack of such an anomaly in regional and global assessments of the flexural bulge outboard of subduction zones would therefore require a coincidence – that there is a cryptic advective extraction term (in addition to that associated with ridge-flank circulation in general) that exactly accounts for excess heat output associated with widespread serpentinization. This could occur, but it seems unlikely, so I think that the burden of proof is on those advocating that such processes occur – show us the data.

In the present study, the emphasis is on B chemistry and isotopics, but the modeling also illustrates the thermal conundrum – Figure 4c would be associated with a significant seafloor heat flux anomaly. Where are the data to support the occurrence of this process, either as single-pass or multi-pass circulation?

I have some additional questions about the simulation configuration. The simulation shows, essentially, a single case with a fixed set of properties – it is not clear how the results depend on parameters like the fault damage-zone width or crustal permeability distribution (not just individual values assigned to large regions, but what is the nature of the network, which is important for reactive transport modeling). These topics are covered, to a greater extent, in Titarenko and McCaig (2015), but there is a limited set of parameter variations. I'm skeptical about quantifying changes in rock properties with temperature/time, as we lack key observational data to supported selected relations (selected relations are not unreasonable, merely poorly constrained), and there are no examinations of more complex distributions or tests of sensitivity to properties or geometries in the present study. For example, what if the fault zone is not perfectly vertical or soles at depth, so that buoyancy helps to drive fluid into the hanging wall? What is the influence of the specific surface area of reaction?

The simulations lack sedimentary cover, which would both help to trap heat and limit inflow/outflow (and thus likely promote multi-pass rather than single pass-flow). And the simulations lack

exothermic heating associated with rock alteration, which would likely have a positive but transient feedback, as previously reacted rock generates successively less heat with additional W/R interaction.

None of this means that the modeling is wrong, but I think it might be best viewed as a means for illustrating the key issues, as a thought problem showing (quantitatively and with a specific example) how multiple-pass simulation could differ from single-pass simulation. It is difficult to show, with modeling, that something cannot occur – this is often a case of affirming the consequent. The results of the simulations depend, to a great extent, in assumptions necessary to create them.

On this basis, I might present the simulations more as a means to illustrate a fundamental difference between the influence of two end-member flow regimes – this provides a strong argument for getting the key observations to determine what kind of hydrogeologic processes occur in the field. It does not undermine or contradict the stated interpretations, but it moderates somewhat the emphasis by which one can assert that the simulations favor a specific result. And the authors can still emphasize that the observational data are more consistent with multi-pass flow.

In summary, I recommend that this paper be published. The simulations are plausible. I recommend that the text describing model results be modified (modestly) to emphasize that the simulations are especially helpful for showing how the hydrogeology should matter. I would lean more strongly on the observational data (petrology, alteration, structural geology) for making the case favored by the authors. That seems to be a strong case, based mainly on mass balance and consideration of isotopic fractionation. I don't think that this approach weakens the arguments – the paper may convince more people because it won't overemphasize the numerical results as being as conclusive as the observational data.

Best wishes, Andy Fisher

McCaig et al.

Reviewer's comments and editorial comments – response

Editorial comments:

P1/2: We have changed the title and shortened the abstract to 149 words

Hopefully Supplementary Figures are now referred to correctly throughout

The list on P18 has been changed to continuous text

A Code Availability section has been added

REVIEWERS' COMMENTS:

Reviewer #1 (Remarks to the Author):

Overall, I think this paper is acceptable for publication. The authors have addressed reviews adequately in my opinion. The text has been improved and I feel laced with sufficient caveats that their main point is credible – namely that in most instances (where hydrothermal circulation is limited), it is unlikely that there will be substantive addition of B to the upper mantle portions of most subducting slabs. Rather, it would be likely that B enrichments may be concentrated in the oceanic crustal section of such slabs. They allow that additional sources of 'excess' B observed in many arc magma suites could include slab domains that have been strongly influenced by penetration of hydrothermal fluids (e.g., fracture zones), or tectonically eroded frontal arc materials that have partially equilibrated with seawater.

Note the following comments:

p. 6, l. 20 typo 'born' -> boron ***done, together with other typos below***

l. 22 Is your 'global average' based just on the few samples you analyzed? Maybe you could define what this really means? The real question is 'what does such an average really represent?'

The word "global" has been replaced by "overall". The uncertainties in this estimate are clarified on P9 under the heading "Boron uptake in the lower oceanic crust and upper mantle". In table 1 we have added standard deviations for the two averages we quote

p. 7 l. 12 given this caveat, what are estimated uncertainties on the LA-ICP data? Is this trivial or significant? Perhaps indicate approximate error bars on Fig. 3b.

We have added the analytical uncertainty (which is low) to the caption of Fig. 3

p. 10 l. 10 typo 'ration' -> ratio

p. 14 l. 8 typo '1km' -> 1 km

l. 10 typo 'seem' -> seen

Re: methods – I am curious as to why the B isotopic measurements were not made in Pisa (since the B was extracted there)? I would expect that the analytical error by TIMS method would be smaller than that obtained at Bristol.

At the time when these analyses were performed the TIMS at Pisa was not performing well. (it would not be appropriate to mention this as I understand the problems have been resolved). The ICPMS method was established for this project, and we have shown good reproducibility on standards in the supporting material at the concentration levels in our sample. The Pisa laboratory is still the best place to do the complex chemistry required for extraction of boron from silicates

Re: modeling – Fig. 4 seems to represent multipass hydrothermal circulation (see caption), wherein presumably there is repeated (continuous?) infiltration of fluid (seawater?). Panel [d] show depths of B-enrichment fronts as a function of time (up to ca. 0.5 Ma). According to this model, it would appear that B-enrichment is largely confined to the oceanic crust.

Accordingly, for your interpretation, it appears that you consider hydrothermal circulation to be generally confined to such short duration. I may have missed it, but perhaps you should explicitly state this – maybe in the caption.

Also, for comparison, it would be instructive to show comparable model outcomes for a long-lived fracture zone like Amlia (Aleutians), and for forearc ultramafic debris as proposed for S. Sandwich subduction zone. I expect this might be feasible without too much effort, and it would certainly help readers visualize differences associated with these 'end member' scenarios.

The model we show in Fig. 4 was designed to investigate hydrothermal circulation in gabbros at Hess Deep, and did not extend into the mantle. The duration matches the time available at Hess Deep. We have sought to clarify this in the text on P12. Dr Titarenko, who performed the modelling, is now working in a completely different research area, and further modelling for this paper is not realistic.

William Leeman

Reviewer #2 (Remarks to the Author):

This second version of the paper "No significant boron in the hydrated mantle of subducting slabs" has improved in clarity. The authors properly addressed the reviewers' comments, providing more clear arguments supporting the conclusions. The main message is now clear, provocative and suitable for the Journal. Not being an expert, I can't fully judge the modelling part of the work.

However, some queries I raised to the first version of this manuscript still need clarification. The paper states that the serpentized slab mantle is not sufficiently B and ¹¹B-rich to affect B release to arcs. An alternative B source for subduction fluids must be found, and the authors propose the subduction-erosion of serpentized forearc mantle. I do agree, though I think this holds true for present day subduction of the Pacific fast-spreading ocean, whose thick crustal gabbroic layer consumes B from infiltrating seawater. This conclusion may not apply to past subduction of slow-spreading oceans with different lithospheric structure. Therefore, I agree that present day slab serpentinites are B-free, but I would be cautious for past subduction systems. As conveyed in the manuscript, this conclusion applies to all subducting slabs, and I suggest smoothing this aspect in the title (e.g. No significant boron in the hydrated mantle of present-day subducting slabs) and in the text.

We have changed the title to say “most subducting slabs”

Specific comments

1. Abstract. Last sentence "We conclude that only in unusual circumstances can the upper mantle of subducting slabs be a source of boron enrichment in arc volcanics, or deep mantle recycling". As explained above, the statement that serpentized slab mantle is B-free is fine for the present case-study, and it must be mentioned this may not apply to past subduction systems, as suggested by many ophiolitic slab-serpentinites. This sentence may change to: We conclude that only in unusual circumstances can the upper mantle of fast-spreading oceans undergoing present-day subduction be a source of boron enrichment in arc volcanics, or deep mantle recycling.

We have cut the abstract down to 150 words as requested. There is no space to qualify the statement by referring to slow spread lithosphere and ophiolite-slab serpentinites. However we have changed the wording to say “We conclude that hydrated mantle in subducting slabs will only rarely contribute to boron enrichment in arc volcanics, or to deep mantle recycling.” By using the future tense we have hopefully restricted the argument to current and ongoing subduction (which is overwhelmingly fast spread crust) , and not past subduction of slow spread crust

2. Background section, page 4. Sentence: "typical of the upper mantle in relatively intact oceanic mantle entering subduction zones". I apologize, but it still is unclear to me what this intact mantle is. I think it should be clearly stated that this mantle occurs in present-day fast spreading plates.

We have added several sentences of clarification to this section

3. Mineral Data section, page 8. The data suggest that part of the B gained during initial background alteration is back-released to fluids during formation of the green alteration, and that ¹¹B fractionates into the fluid. Can this new ¹¹B-bearing fluid affect serpentization of the mantle rocks underneath?

On page 9 we have clarified our opinion that in the case of Hess Deep, the boron released during the overprinting alteration is released back into the ocean through venting

4. Mineral Data section, page 9. Sentence" Our results show that the olivine-rich lower oceanic crust can be significantly enriched in [B], and have high values of $\delta^{11}\text{B}$, in the vicinity of fault zones, with much higher values of [B] than seen so far in ophiolites". Figure 3a shows that the B and ^{11}B of the Oman gabbros are comparable to those presented in this paper, with the only exception of two enriched samples. This should be better clarified.

Here we disagree with the reviewer. The Oman gabbros are significantly lower in [B], and the two datasets do not overlap. The $\delta^{11}\text{B}$ values do overlap. E have tried to clarify this.

Figure 1. The labels AM15, AM16, overprint, cpx vein are not clearly visible. In the caption it should be explained that the sample from Fig. 1a is divided into part AM15, showing preferential development of the background alteration and into part AM16 characterized by preferential development of the green alteration. Correct (cpyx) into (cpx)

We have improved visibility of labels on this and other figure, and added to the captions

Reviewer #3 (Remarks to the Author):

Review of McCaig et al. (2018), "No significant boron in the hydrated mantle of subducting slabs"

Review by Andrew T. Fisher, 8/26/18

This is a study of rock petrology/alteration and subseafloor hydrogeology, examining if and how [B] and B isotopic values in crust and mantle rocks might be influenced by fluid circulation. The key petrographic comparison is between rock samples collected at multiple field sites on land and on/below the seafloor, whereas the key hydrogeologic comparison is between single-pass and multiple-pass hydrothermal circulation.

I was not a reviewer of the original submission, but was asked to comment on the revision wrt the hydrogeologic modeling. I'll note up front that I share the author's skepticism about single-pass circulation being a ubiquitous and pervasive process along flexural faults at subduction zones. My skepticism extends to multipass circulation in flexural faults. My biggest concern is with the lack of a consistent thermal signal that should be associated with this process, if it were to occur – it would generate a readily detectable seafloor heat flux anomaly. The lack of such an anomaly in regional and global assessments of the flexural bulge outboard of subduction zones would therefore require a coincidence – that there is a cryptic advective extraction term (in addition to that associated with ridge-flank circulation in general) that exactly accounts for excess heat output associated with widespread serpentinization. This could occur, but it seems unlikely, so I think that the burden of proof is on those advocating that such processes occur – show us the data.

We note that the reviewer does not in the end recommend that we address these issues of heat flow around bend faults, which is really beyond the scope of our paper.

In the present study, the emphasis is on B chemistry and isotopics, but the modeling also illustrates

the thermal conundrum – Figure 4c would be associated with a significant seafloor heat flux anomaly. Where are the data to support the occurrence of this process, either as single-pass or multi-pass circulation?

We have emphasised that our model applies primarily to Hess Deep, and have included a statement that our model (with the high temperature hydrothermal assemblages) predicts undiscovered seafloor vents west of the study area on P13.

I have some additional questions about the simulation configuration. The simulation shows, essentially, a single case with a fixed set of properties – it is not clear how the results depend on parameters like the fault damage-zone width or crustal permeability distribution (not just individual values assigned to large regions, but what is the nature of the network, which is important for reactive transport modeling). These topics are covered, to a greater extent, in Titarenko and McCaig (2015), but there is a limited set of parameter variations. I'm skeptical about quantifying changes in rock properties with temperature/time, as we lack key observational data to supported selected relations (selected relations are not unreasonable, merely poorly constrained), and there are no examinations of more complex distributions or tests of sensitivity to properties or geometries in the present study. For example, what if the fault zone is not perfectly vertical or soles at depth, so that buoyancy helps to drive fluid into the hanging wall? What is the influence of the specific surface area of reaction?

We have only limited model results, but given the interest shown by this reviewer, we have added a comparison between 1km and 2 km fault slots in Supplementary figure 4, and a video of the evolution of temperature in Supplementary Video 1

The simulations lack sedimentary cover, which would both help to trap heat and limit inflow/outflow (and thus likely promote multi-pass rather than single pass-flow). And the simulations lack exothermic heating associated with rock alteration, which would likely have a positive but transient feedback, as previously reacted rock generates successively less heat with additional W/R interaction.

None of this means that the modeling is wrong, but I think it might be best viewed as a means for illustrating the key issues, as a thought problem showing (quantitatively and with a specific example) how multiple-pass simulation could differ from single-pass simulation. It is difficult to show, with modeling, that something cannot occur – this is often a case of affirming the consequent. The results of the simulations depend, to a great extent, in assumptions necessary to create them.

We clarified what we mean y single pass and multipass on P11

On this basis, I might present the simulations more as a means to illustrate a fundamental difference between the influence of two end-member flow regimes – this provides a strong argument for getting the key observations to determine what kind of hydrogeologic processes occur in the field. It does not undermine or contradict the stated interpretations, but it moderates somewhat the

emphasis by which one can assert that the simulations favor a specific result. And the authors can still emphasize that the observational data are more consistent with multi-pass flow.

We agree that our model should be seen as illustrative rather than definitive, and have added words on P12/13 to this effect.

In summary, I recommend that this paper be published. The simulations are plausible. I recommend that the text describing model results be modified (modestly) to emphasize that the simulations are especially helpful for showing how the hydrogeology should matter. I would lean more strongly on the observational data (petrology, alteration, structural geology) for making the case favored by the authors. That seems to be a strong case, based mainly on mass balance and consideration of isotopic fractionation. I don't think that this approach weakens the arguments – the paper may convince more people because it won't overemphasize the numerical results as being as conclusive as the observational data.

Best wishes, Andy Fisher